# WAVE-PARTICLE (CONTINUOUS–DISCRETE) DUALISTIC VISUAL TOKENIZATION FOR UNIFIED UNDERSTANDING AND GENERATION

## ABSTRACT

The unification of understanding and generation within a single multi-modal large model (MLLM) remains one significant challenge, largely due to the dichotomy between continuous and discrete visual tokenizations. Continuous tokenizer (CT) achieves strong performance by bridging multiple independently-trained understanding modules and generation modules, but suffers from complex multi-stage pipelines and substantial engineering overhead. Conversely, discrete tokenizers (DT) offer a conceptually elegant idea by quantizing each image into a primitive, but inevitably leading to information loss and performance degradation. To resolve this tension, we question the binary choice between CT and DT, inspired by the wave-particle duality of light, and propose the Continuous-Discrete Dualistic Visual Tokenizer (CDD-VT). We treat visual data as a flexible composition of image primitives derived from quantized codebooks, with the crucial insight that the primitive number assigned to each visual sample is adaptively determined according to its complexity: simple instances use a few primitives, emulating discrete tokenization, while complex instances use many, approximating continuous tokenization. Two core components are designed: Diverse Quantitative Primitives, which encourage primitives orthogonality to better populate information space, and Dynamic Primitive Allocator, which assesses sample complexity to determine the optimal set of primitives. Extensive experiments on reconstruction, retrieval and classification demonstrate that CDD-VT achieves superior performance over to specialized CT and DT, effectively getting strong result within a concise and scalable MLLM.

## 1 INTRODUCTION

In recent years, the rapid development of large-scale models (LMs) has swept through the artificial intelligence community, reshaping productivity everywhere and demonstrating transformative impacts across a broad spectrum of downstream applications. Generally speaking, the capabilities of LMs can be roughly divided into two fundamental groups: understanding and generation. This dichotomy has given rise to several architectural branches, *e.g.*, Large Language Models (LLMs) that excel in text understanding and generation, diffusion models (DFMs) that pioneer high-fidelity image generation, and multimodal large language models (MLLMs) that integrate image-text understanding.

In NLP, LLMs have unified understanding and generation, but in the multimodal domain, DFMs and MLLMs remain largely separate, leaving multimodal unity an underexplored frontier. Vision tokenization is crucial for this, enabling LMs to process image-text inputs and generate image-text outputs, as depicted in Figure 1. Recent studies fall into two groups, continuous tokenizers (CT) (Deng et al., 2025; Dong et al., 2023; Ge et al., 2023; Li et al., 2024b) and discrete tokenizers (DT) (Wu et al., 2025; 2024b; Ma et al., 2025). The CT paradigm typically employs an encoder-projector pipeline to map visual inputs into embeddings, which are subsequently aligned with LLMs for understanding. Conditioned on the output tokens generated by LLMs, an external DFM adapter is invoked to enable visual generation. By reusing existing state-of-the-art MLLMs and DFMs, CT attains strong performance. However, its multi-stage and multi-round workflow is relatively complex, often relying on numerous ad-hoc tricks. Given that existing LM pipelines and systems are already highly optimized for next-token prediction, maintaining an additional adapter to support DFMs entails substantial engineering overhead. On the other hand, the DT paradigm discretizes visual data into

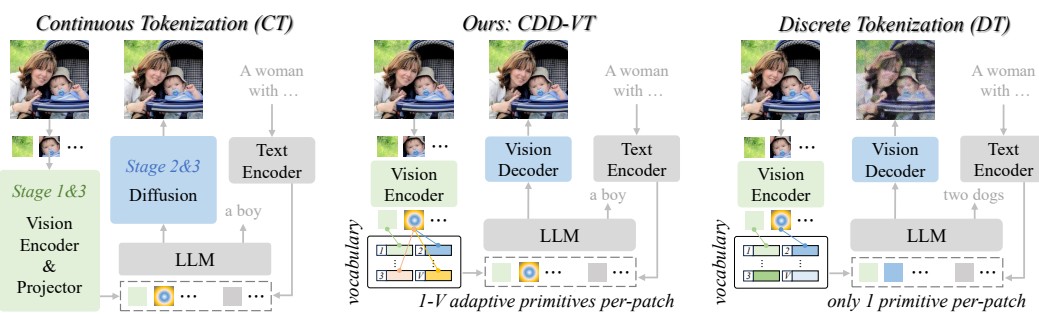

Figure 1: **Tokenization Comparisons.** *Continuous Tokenizers (CT)*: strong performance, but one complex, multi-stage workflow. *Discrete Tokenizers (DT)*: a concise-unified workflow, but poor results. Our **CDD-VT** as the continuous-discrete dualistic quantization, achieves remarkable results in elegant unity of understanding and generation.

tokens via vector quantization (VQ), treating visual and textual modalities as equivalent and thereby unifying generation and understanding within a single MLLM. While CT offers conceptual simplicity and VQ techniques have undergone multiple advancements (van den Oord et al., 2017; Esser et al., 2021; Yu et al., 2021; Lee et al., 2022), the VQ codebook inevitably discards significant amount of information, resulting in inferior performance.

To address the inherent tension: maintaining a concise-scalable framework while pursuing strong performance, we examine a fundamental question: must visual tokenization be limited to the binary choice between discrete or continuous? *Inspired by the famous wave–particle duality of light in physics, namely light can be viewed as both a continuous wave and a discrete particle, we propose a continuous–discrete dualistic visual tokenizer (CDD-VT) to unify understanding-generation.*

For the sake of simplicity and scalability, CDD-VT employs VQ techniques to define image primitives, serving as the visual equivalent of sub-word morphemes in NLP. For strong performance, since these image primitives are somewhat orthogonal, CDD-VT considers one single visual data as a composition of multiple image primitives, analogous to how some sub-word morphemes in NLP can be concatenated to form one word. This design is guided by two key insights: (A) Combining sufficient primitives enables the preservation of visual data with negligible information loss, thereby approximating continuous tokenization. (B) Rather than assigning one or a fixed number of primitives, their allocation should be adaptively determined based on complexity and granularity of each visual sample. Building upon these insights, CDD-VT employs a single or a few primitives for simple visual instances, functioning as discrete tokenization; whereas for complex visual instances, many-up to all-primitives is utilized, serving as continuous tokenization.

Concretely, CDD-VT is composed of a vision encoder, a text encoder, a vision decoder, and two core components, namely **Diverse Quantitative Primitives (DQP)** and **Dynamic Primitive Allocator (DPA)**. *Tokenization stage.* The image or text is partitioned into patches or sub-words, and then fed into the vision or text encoder for latent embeddings. DQP statistics the data distribution to construct a set of diverse quantitative primitives. DPA assesses information complexity of each sample, and dynamically quantifies patch embeddings into combinations covering different numbers of primitives, which are finally aligned with text embeddings. *De-tokenization stage.* The vision decoder reconstructs the image from these adaptively allocated primitives. To pursuit orthogonal image primitives that can sufficiently populate the information space and thus mitigate information loss, DQP follows the principle of codebook quantization while updating it to encourage diversity regularization between multiple sub-codebooks. To realize adaptive primitive allocator, DPA uses the reconstructing difficulty of each sample during training as a prior of information complexity. It then ranks primitives according to correlation and dynamically selects the optimal number of primitives, minimizing information loss during discretization.

On tasks such as image classification, image-text retrieval, image reconstruction across three classic benchmarks, we carry out extensive experiments to demonstrate the superiority of CDD-VT, being better or comparable to discrete tokenizers and continuous tokenizers. Concretely, on ImageNet, CDD-VT records an impressive 0.31 reconstruction FID and 70.5% zero-shot Top-1 accuracy at 256×256 resolution.

## 2 RELATED WORK

**Discrete Image Tokenizers** convert visual signals into a sequence of discrete tokens. The pioneering VQVAE (van den Oord et al., 2017) first introduced to quantize latent embeddings using a finite codebook, where each embedding is replaced by its nearest discrete primitive. This was later improved upon by VQGAN (Esser et al., 2021), which integrated a GAN loss to enhance reconstruction quality.

Recent studies, including LlamaGen (Sun et al., 2024), VAR (Tian et al., 2024), and UniTok (Ma et al., 2025), have demonstrated the significant potential of discrete tokenizers for autoregressive (AR) image generation. However, due to the significant information loss induced by quantization, the performance upper bound of AR methods may be limited, as evidenced in Wu et al. (2025); Xie et al. (2024). In our work, we address this limitation by using adaptive granularity quantization to dynamically generate finer-grained tokens, which substantially improves reconstruction quality.

**Continuous Image Tokenizers** map image patches to continuous latent tokens, providing more natural embeddings with typically less information loss. A common design employs an encoder with an MLP or linear projector to compress high-dimensional images into a low-dimensional latent space. However, their probabilistic nature often requires complex components, such as diffusion heads, to model the distribution effectively. Recent approaches, including MAR (Li et al., 2024a), generate continuous tokens autoregressively using a diffusion loss, while Fluid (Fan et al., 2024) scales this paradigm to large-scale text-to-image generation. MetaMorph (Tong et al., 2024) instead directly predicts continuous SigLIP (Zhai et al., 2023) embeddings. FAR (Yu et al., 2025) further satisfies autoregressive causality constraints, preserves spatial locality, and improves sampling efficiency, making it a practical choice for integrating continuous tokenization into AR training.

**Unified Multimodal Understanding and Generation Models** aim to consolidate both understanding and generation capabilities within a single architecture across multiple modalities. Two primary approaches exist for visual content generation in these models. The first integrates continuous visual tokenizers with diffusion models, such as Stable Diffusion (Rombach et al., 2022), to generate high-fidelity images (Dong et al., 2023; Ge et al., 2023; Li et al., 2024b; Sun et al., 2023). The second approach focuses on effectively tokenizing images for autoregressive generation. Some methods utilize discrete tokenizers to convert visual inputs into discrete tokens, maintaining a purely autoregressive paradigm (Wu et al., 2024b;a; Wang et al., 2024; Ma et al., 2025; Zhao et al., 2025), while others incorporate diffusion elements, either for decoding continuous tokens (Tong et al., 2024) or as part of the token prediction process itself (Li et al., 2024a). For evaluation, we compare our method against several representative works from these categories.

## 3 METHOD: CDD-VT

For the unification of understanding and generation, the key is to maintain concise-scalable framework while pursuing strong performance. Hence, we propose a Continuous-Discrete Dualistic Visual Tokenizer (CDD-VT). In Sec. 3.1, we first outline formulation and framework; in Sec. 3.2, we pursue orthogonal image primitives; in Sec. 3.3, we adaptively allocate the optimal set of primitives by information complexity; in Sec. 3.4, we summarize training stages of CDD-VT and all optimization objectives. An overview of the framework is presented in Figure 2.

### 3.1 FORMULATION & FRAMEWORK

Following the classic consensus (van den Oord et al., 2017), our CDD-VT adopts a robust tokenization-quantization-detokenization pipeline, corresponding to an image encoder ($\mathcal{E}_\text{I}$), a text encoder ($\mathcal{E}_\text{T}$), and a decoder ($\mathcal{D}$). Here, codebook quantization covers two core components: the **D**iverse **Q**uantitative **P**rimitives ($\mathcal{Q}_\text{DQP}$) and the **D**ynamic **P**rimitive **A**llocator ($\mathcal{Q}_\text{DPA}$).

During **tokenization**, for one input image $\mathbf{I} \in \mathbb{R}^{H' \times W' \times C}$, we first partition it into patches, and then pass patches through $\mathcal{E}_\text{I}$ into latent embeddings $\mathbf{Z}_\text{I} = \mathcal{E}_\text{I}(\mathbf{I}) \in \mathbb{R}^{HW \times D}$, where $D$ is the embedding dimension and $H/W$ denotes the downsampling of the spatial dimensions $H'/W'$. For one input text $\mathbf{T} \in \mathbb{R}^N$ of $N$ words, we obtain its latent embeddings by $\mathbf{Z}_\text{T} = \mathcal{E}_\text{T}(\mathbf{T}) \in \mathbb{R}^{N \times D}$.

During **quantization**, $\mathcal{Q}_\text{DQP}$ constructs a set of diverse quantitative primitives $\{\mathbf{c}_i\}_{i=1}^V$, where $V$ is the total number of primitives. For each image embeddings $\mathbf{Z}_\text{I}$, $\mathcal{Q}_\text{DPA}$ assesses information complexity

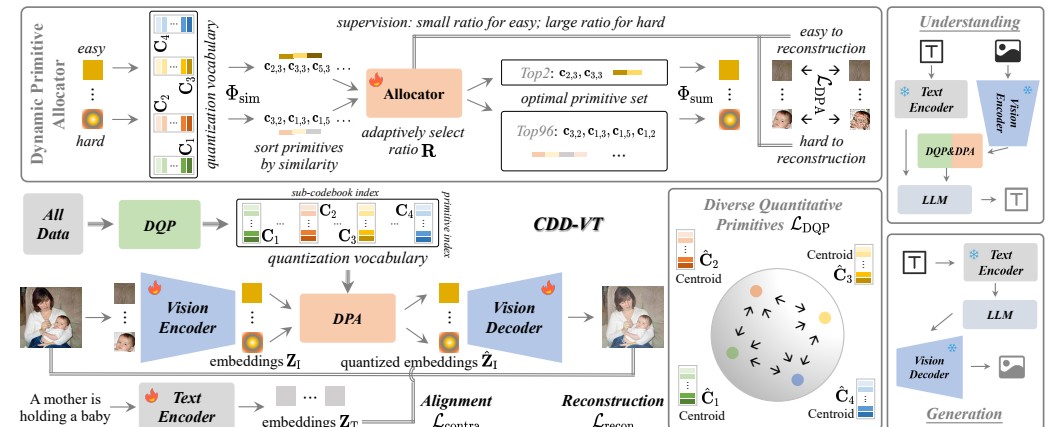

Figure 2: **Framework Overview.** Our continuous–discrete dualistic visual tokenizer (CDD-VT) consists of an image encoder, a text encoder, an image decoder, and two core components: Diverse Quantitative Primitives (DQP) and Dynamic Primitive Allocator (DPA). DQP encourages primitives orthogonality to better populate information space, while DPA assesses sample complexity to determine the optimal primitive set for each patch. For understanding, we calculate cosine similarity between text embeddings and quantified image embeddings. For generation, we feed text through the pipeline of encoder, LLM, and vision decoder.

of its $H \times W$ patches, and dynamically quantifies each patch embeddings into an optimal set $\{\mathbf{C}_i\}_{i=1}^{O} \in \{\mathbf{c}_i\}_{i=1}^{V}$. This operation maps $\mathbf{Z}_\mathrm{I}$ into $\hat{\mathbf{Z}}_\mathrm{I}$, which is aligned with text embeddings $\mathbf{Z}_\mathrm{T}$.

During **de-tokenization**, $\mathcal{D}$ reconstructs image $\hat{\mathbf{I}}$ from $\hat{\mathbf{Z}}_\mathrm{I}$, and the overall formulation is:

$$\hat{\mathbf{I}} = \mathcal{D}(\hat{\mathbf{Z}}_\mathrm{I}) = \mathcal{D}(\mathcal{Q}_\mathrm{DPA}(\mathbf{Z}_\mathrm{I})) = \mathcal{D}(\mathcal{Q}_\mathrm{DPA}(\mathcal{E}_\mathrm{I}(\mathbf{I}))) \quad \{\mathbf{C}_i\}_{i=1}^{C} = \mathcal{Q}_\mathrm{DQP}(\mathbf{I}). \tag{1}$$

**Multimodal Autoregression.** We extend CDD-VT to Multimodal Large Language Models (MLLMs) by unifying vision and language. For image understanding, visual patches and text tokens are modeled with a universal next-token loss. Rather than training a discrete visual codebook from scratch, CDD-VT decomposes images into compositional primitives and maps their continuous embeddings into the MLLM token space via a lightweight MLP projector. For image generation, the model takes text as conditioning and predicts the next continuous visual tokens in an autoregressive manner. Following MAR (Li et al., 2024a), we adoptprogressively denoises tokens in a "next-set" autoregressive manner, starting from masked tokens.

### 3.2 DIVERSE QUANTITATIVE PRIMITIVES

For the primitive quantification, one typical solution is codebook construction (van den Oord et al., 2017; Esser et al., 2021), *i.e.*, modeling data distribution to obtain fixed primitive embeddings that serve as the basic units for decomposing continuous embeddings. The challenge lies in minimizing information loss. Here, we encourage primitive embeddings to present one more diverse distribution, thereby achieving more complete coverage of the continuous information space.

To maximize primitive diversity, we employ a multi-element, multi-dimensional codebook structure, akin to the multi-head attention. Denoting one primitive as $\mathbf{c} \in \mathbb{R}^{D'}$, and the vocabulary size as $V'$, we combine $V'$ primitives into a sub-codebook $\mathbf{C}_j = \{\mathbf{c}_{i,j}\}_{i=1}^{V'} \in \mathbb{R}^{V' \times D'}$; then concatenate $M$ sub-codebooks along the embedding dimension as the whole codebook $\mathbf{C} = \{\mathbf{C}_j\}_{j=1}^{M} \in \mathbb{R}^{V' \times D' \times M}$, where $V' \times M = V$, $D' \times M = D$. Image embeddings $\mathbf{Z}_\mathrm{I} \in \mathbb{R}^{HW \times D}$ are subsequently split into $M$ chunks, i.e., $\{\mathbf{Z}_\mathrm{I1}, \dots, \mathbf{Z}_\mathrm{IM}\} \in \mathbb{R}^{HW \times D'}$, each quantized into a combination of primitives.

During codebook optimization, we explicitly encourage primitive diversity by minimizing the pairwise cosine similarity ($\Phi_\mathrm{sim}$) between sub-codebook centroids. This addresses potential primitive homogenization, a common challenge in codebooks, and results in sub-codebook regularization:

$$\mathcal{L}_\mathrm{DQP} = \frac{2}{M(M-1)} \sum_{1 \leq j < k \leq M} \Phi_\mathrm{sim}(\hat{\mathbf{C}}_j, \hat{\mathbf{C}}_k), \quad \hat{\mathbf{C}}_j = \frac{1}{v} \sum_{i=1}^{v} \mathbf{c}_{i,j}, \tag{2}$$

where $M$ is the sub-codebook number, $\mathbf{c}_{i,j}$ is i-th primitive from j-th sub-codebook, and $\hat{\mathbf{C}}_i$ denotes the centroid of the $i$-th sub-codebook. Minimizing $\mathcal{L}_{\text{DQP}}$ compels sub-codebooks to occupy distinct regions in the information space, enhancing the coverage of resultant discrete primitive set.

## 3.3 Dynamic Primitive Allocator

After establishing diverse quantitative primitives, the next key step is effectively allocating an optimal set to each patch embedding, *i.e.*, determining both the primitive number to use and which ones to select. Traditionally, discrete tokenizers map each patch embedding to its single nearest codebook primitive, which inevitably incurs information loss. In contrast, we dynamically assess each patch's information complexity via reconstruction difficulty, rank primitives by correlation, and select an optimal number, hence minimizing information loss and approximating continuous tokenizers.

**The Optimal Set of Primitives.** For one input image embeddings $\mathbf{Z}_{\text{I}}$, to choose optimal primitives from $j$-th sub-codebook $\mathbf{C}_j$, we first calculate the correlation ($\Phi_{\text{sim}}$) between each patch and all primitives from $\mathbf{C}_j$, then use one **allocator** $\mathcal{A}$ to determine the **allocation ratio** $\mathbf{R} \in (0, 1]$, *i.e.*, the exact number of how many primitives to keep. Here, a higher $\mathbf{R}$ value signifies greater complexity, and needs more primitives to maintain information. And to simplify the calculation, we set the maximum selection number to Top-$K$ ($K$ a hyperparameter, $\Phi_{\text{top}}$). The allocator consists of two 1D convolutional layers followed by a sigmoid function. Formally,

$$\hat{\mathbf{Z}}_{\text{I}j} = \Phi_{\text{sum}}(\Phi_{\text{top}}(\Phi_{\text{sim}}(\mathbf{Z}_{\text{I}}, \mathbf{C}_j), \mathbf{R}, K)) \quad \mathbf{R} = \mathcal{A}(\mathbf{Z}_{\text{I}}). \tag{3}$$

The resulting quantized embeddings, namely $\hat{\mathbf{Z}}_{\text{I}j}$, are computed as a weighted sum $\Phi_{\text{sum}}$ of adaptively selected primitives, with the weight derived from their correlation.

**Measurement of Information Complexity.** We here follow one intuitive but reliable prior, *i.e.*, for a given patch, larger VAE reconstruction errors suggest higher information complexity, characterized by greater resolution, richer color, more diverse textures, and vice versa. Such a prior is natural because regions with higher information complexity are more prone to information loss during tokenization-quantization-detokenization. Concretely, reconstruction errors are calculated by the squared L2-norm between quantized patch embeddings $\hat{\mathbf{Z}}_{\text{I}}$ and original embeddings $\mathbf{Z}_{\text{I}}$. Then we normalize squared L2-norm into a range of $\frac{1}{V'}$-1, as the optimization guide for allocation ratio $\mathbf{R}$:

$$\mathcal{L}_{\text{DPA}} = \text{MSE}(\mathbf{R}, \mathbf{R}^\star), \quad \mathbf{R}^\star = \Phi_{\text{norm}}(\|\hat{\mathbf{Z}}_{\text{I}} - \mathbf{Z}_{\text{I}}\|_2^2) \tag{4}$$

Hence, we employ a single or a few primitives for simple patches, functioning as discrete tokenization; whereas for complex patches, many-up to all-primitives, is used, serving as continuous tokenization.

**Warm-Up Training.** Since DPA is guided by reconstruction errors, which typically stabilize after approximately the first quarter of the training stage (see Figure 4), we here employ one warm-up strategy. During the initial quarter phase of training, both DQP and DPA remain inactive, *i.e.*, the quantization stage operates as discrete, selecting only the single nearest primitive for each patch embedding. Subsequently, following the warm-up period, both DQP and DPA are fully activated for the remainder of training, enabling dynamic primitive allocation and diversity encouragement.

## 3.4 Training & Evaluation

**Training.** Building on VQGAN (Esser et al., 2021), our reconstruction objective $\mathcal{L}_{\text{recon}}$ combines reconstruction, commitment, perceptual, and adversarial losses. Beyond these, our final training objective also includes losses for our specific innovations and for image-text alignment. The total loss is thus defined as the $\lambda$-weighted sum of all these components:

$$\mathcal{L}_{\text{total}} = \mathcal{L}_{\text{recon}} + \lambda_{\text{contra}}\mathcal{L}_{\text{contra}} + \lambda_{\text{DQP}}\mathcal{L}_{\text{DQP}} + \lambda_{\text{DPA}}\mathcal{L}_{\text{DPA}}. \tag{5}$$

$\mathcal{L}_{\text{contra}}$ is the contrastive loss computed via InfoNCE (Radford et al., 2021) to enforce alignment between text embeddings $\mathbf{Z}_{\text{T}}$ and quantized image embeddings $\hat{\mathbf{Z}}_{\text{I}}$.

**Evaluation.** For understanding tasks, *e.g.*, classification and retrieval, we use trained (image encoder & quantitative primitives) and text encoder separately to map images and text to embeddings, that is, $\hat{\mathbf{Z}}_{\text{I}} = \mathcal{Q}_{\text{DPA}}(\mathcal{E}_{\text{I}}(\mathbf{I}))$ and $\mathbf{Z}_{\text{T}} = \mathcal{E}_{\text{T}}(\mathbf{T})$. Then we calculate cosine similarity between image-text embeddings, and sort for prediction. Here, a set of predefined class names is regarded as text for usage. For reconstruction tasks, the image encoder $\mathcal{E}_{\text{I}}$ takes into $\mathbf{I}$, and passes it through the quantitative primitives and decoder to get the reconstructed image $\hat{\mathbf{I}} = \mathcal{D}(\mathcal{Q}_{\text{DPA}}(\mathcal{E}_{\text{I}}(\mathbf{I})))$.

Table 1: **Image Reconstruction and Zero-Shot Image Classification on ImageNet 50K Validation Set.** [†] indicates model using CLIP pre-trained weights for initialization. [*] denotes our reproduction.

| Method | Data | #Tokens | Res. | Codebook Size | Accuracy | rFID↓ | PSNR↑ | LPIPS -VGG↓ |
|---|---|---|---|---|---|---|---|---|
| *Continuous Tokenizer (CT)* | | | | | | | | |
| EVA02-B/16 | Merged-2B | / | 224 | / | 74.7 | / | / | / |
| CLIP-L/14 | WIT400M | / | 224 | / | 75.5 | / | / | / |
| MetaCLIP-L/14 | MetaCLIP400M | / | 224 | / | 76.2 | / | / | / |
| *Discrete Tokenizer (DT)* | | | | | | | | |
| VQGAN | ImageNet1k | 256 | 256 | 16384 | - | 4.98 | 20.00 | 0.2843 |
| SD-VQGAN | OpenImages | 256 | 256 | 16384 | - | 5.15 | 20.83 | - |
| LlamaGen | ImageNet1k | 256 | 256 | 16384 | - | 2.19 | 20.79 | - |
| TokenFlow[†] | LAION+ COYO700M | 680 | 256 | 32768 | - | 1.37 | 21.41 | - |
| VILA-U[†] | COYO700M | 256 | 256 | 16384 | 73.3 | 1.80 | - | - |
| TiTok-S | MaskGIT-VQGAN | 128 | 256 | 4096 | - | 1.71 | - | - |
| QLIP[†] | DC-1B | - | 256 | - | 74.3 | 3.21 | 23.16 | - |
| Upper Bound | DC-800M | / | 256 | / | 66.8 | / | / | / |
| UniTok[*] | DC-800M | 256 | 256 | 32768 | 65.8 | 0.41 | 23.57 | 0.1440 |
| **CDD-VT** | DC-800M | 256 | 256 | 32768 | 66.5 | 0.37 | 23.93 | 0.1410 |
| Upper Bound | DC-1B-R | / | 256 | / | 70.1 | / | / | / |
| UniTok | DC-1B | 256 | 256 | 32768 | 70.8 | 0.41 | 23.80 | 0.1458 |
| UniTok[*] | DC-1B-R | 256 | 256 | 32768 | 68.6 | 0.41 | 23.90 | 0.1366 |
| **CDD-VT** | DC-1B-R | 256 | 256 | 32768 | 69.8 | 0.36 | 24.51 | 0.1289 |
| UniTok[†] | DC-1B | 256 | 256 | 32768 | 78.6 | 0.38 | - | - |
| UniTok[†*] | DC-1B-R | 256 | 256 | 32768 | 69.9 | 0.35 | 24.40 | 0.1288 |
| **CDD-VT**[†] | DC-1B-R | 256 | 256 | 32768 | 70.5 | 0.31 | 24.95 | 0.1197 |

## 4 EXPERIMENT

**Datasets & Metrics.** For visual reconstruction, we present results on the validation sets of ImageNet-1k and MSCOCO 2014. The quality of reconstructed images is measured by a standard set of metrics: reconstruction-FID (rFID) (Heusel et al., 2017), PSNR, and SSIM (Wang et al., 2004). For zero-shot image-text retrieval, evaluations are conducted on the 5K validation set of MSCOCO 2014 (Lin et al., 2014) and the 1K test set of Flickr30K (Young et al., 2014). Following established practices (Radford et al., 2021; Zhai et al., 2022; 2023), we leverage the widely-used Karpathy splits (Karpathy & Fei-Fei, 2015) for both datasets and measure performance using Recall@K (R@K). Simultaneously, we assess the quality of visual generation using prevalent metrics, including FID, Inception Score (IS) (Salimans et al., 2016), and Precision/Recall (Kynkäänniemi et al., 2019).

**Tokenizer Setup.** We configure the number of sub-codebooks to 8, each containing 4096 primitives. and a local embedding dimension $D'$ of 8-d, resulting in a global embedding dimension $D$ of 64-d, following UniTok (Ma et al., 2025). The encoder and decoder are built upon the ViTamin-L/16 (Chen et al., 2024). We train CDD-VT for one epoch on the public dataset DataComp-1B (DC-1B) (Gadre et al., 2023) consisting of 1.28B image-text pairs[1] All images are resized to a $256 \times 256$ resolution and a local batch size of 64. The learning rates are set to 1e-3 for the tokenizer. For evaluation, we test two initialization settings: CLIP pre-trained weights and default random initialization.

### 4.1 TOKENIZER COMPARISONS

Table 1 first evaluates on the ImageNet 50K validation set (Russakovsky et al., 2015) across classification and reconstruction. As comparisons, we include several continuous tokenizers, including EVA02-B/16 (Fang et al., 2024), CLIP-L/14 (Radford et al., 2021), and MetaCLIP-L/14 (Xu et al., 2023); as well as prominent discrete tokenizers such as VQGAN (Esser et al., 2021), SD-VQGAN (Rombach et al., 2022), Llama-Gen (Sun et al., 2024), TokenFlow (Qu et al., 2025), VILA-U (Wu et al., 2024b), TikTok-S (Yu et al., 2024), QLIP (Zhao et al., 2025), and UniTok (Ma et al., 2025).

---

[1]Due to download issues, 0.8B pairs (DC-800M) were directly obtained. The remaining 0.48B pairs were then resampled to complete the 1.28B dataset, which we denote as DC-1B-R.

Table 2: **Zero-Shot Image Reconstruction**. [†] means initialization from CLIP pretrained weights. [‡] means LlamaGen loads weights trained on ImageNet while other weights are trained from scratch, *i.e.*, MS-COCO, ImageNet-1k are excluded from training data. [*] is our reproduction. All methods are trained with 256 tokens and a downsampling ratio of 16.

| Method | Data | Codebook | MS-COCO 2017 | | | ImageNet-1k | | |
|---|---|---|---|---|---|---|---|---|
| | | Size | rFID↓ | PSNR↑ | SSIM↑ | rFID↓ | PSNR↑ | SSIM↑ |
| LlamaGen[‡] | 70M | 16384 | 8.40 | 20.28 | 0.55 | 2.47 | 20.65 | 0.54 |
| Show-o | 35M | 8192 | 9.26 | 20.90 | 0.59 | 3.50 | 21.34 | 0.59 |
| Cosmos | – | 64000 | 11.97 | 19.22 | 0.48 | 4.57 | 19.93 | 0.49 |
| Open-MAGVIT2-I-PT | 100M | 262144 | 6.76 | 22.31 | 0.65 | 1.67 | 22.70 | 0.64 |
| IBQ | 100M | 262144 | 6.79 | 22.28 | 0.65 | 1.53 | 22.69 | 0.64 |
| TokenFlow[†] | LAION+ COYO700M | 32768 | - | - | - | 1.37 | 21.41 | 0.69 |
| UniTok | DC-1B | 32768 | 4.22 | 23.38 | 0.68 | 0.41 | 23.80 | 0.70 |
| **CDD-VT** | DC-1B-R | 32768 | 3.86 | 24.07 | 0.71 | 0.36 | 24.51 | 0.73 |

Table 3: **Results on Zero-Shot Image-Text Retrieval**. [†] indicates model using pretrained CLIP weights for initialization. [*] denotes our reproduction. Reuslts show that random initialized CDD-VT out performs UniTok and even comparable to CLIP initialized QLIP.

| Method | Res. | Data | Image to Text | | | | | | Text To Image | | | | | |
|---|---|---|---|---|---|---|---|---|---|---|---|---|---|---|
| | | | Flickr30k | | | MSCOCO | | | Flickr30k | | | MSCOCO | | |
| | | | R@1 | R@5 | R@10 | R@1 | R@5 | R@10 | R@1 | R@5 | R@10 | R@1 | R@5 | R@10 |
| *Continuous Tokenizer (CT)* | | | | | | | | | | | | | | |
| EVA02-B/16 | 224 | Merged-2B | 87.9 | 97.8 | 99.2 | 60.0 | 82.7 | 89.1 | 73.9 | 91.1 | 94.7 | 42.2 | 67.5 | 77.1 |
| CLIP-L/14 | 224 | WIT400M | 86.4 | 97.3 | 99.4 | 57.1 | 80.1 | 87.2 | 67.9 | 88.9 | 93.3 | 35.4 | 60.4 | 71.0 |
| *Discrete Tokenizer (DT)* | | | | | | | | | | | | | | |
| QLIP[†] | 256 | DC-1B | 80.4 | 94.0 | 97.0 | 50.6 | 74.1 | 82.2 | 61.8 | 84.5 | 89.9 | 32.9 | 57.2 | 67.6 |
| UniTok | 256 | DC-1B | 79.5 | 95.6 | 97.3 | 52.4 | 76.4 | 84.4 | 60.8 | 84.4 | 90.3 | 34.8 | 59.6 | 69.9 |
| UniTok[*] | 256 | DC-1B-R | 75.8 | 93.3 | 96.6 | 50.9 | 74.5 | 83.4 | 57.9 | 82.8 | 89.0 | 33.2 | 57.6 | 68.2 |
| **CDD-VT** | 256 | DC-1B-R | 78.7 | 93.3 | 96.5 | 53.0 | 75.4 | 83.8 | 59.3 | 82.8 | 89.0 | 34.0 | 58.8 | 69.1 |

CDD-VT demonstrates exceptional reconstruction quality, surpassing all state-of-the-art tokenizers. It achieves an impressive rFID of **0.31** on ImageNet with a $16\times$ downsampling ratio. Moreover, CDD-VT surpasses all other tokenizers in PSNR, achieving on the full Datacomp-1B (including 480M resampled images) and on the 800M subset. These results validate the effectiveness of DQP and DPA's combination. Besides, when expanding training data from 800M to resampled 1B, further reconstruction gains are observed, demonstrating the scalability of CDD-VT.

Regarding image-text aligned understanding, CDD-VT achieves suboptimal, yet competitive, results in zero-shot classification, reaching 70.5% Top-1 accuracy on ImageNet and notably surpassing UniTok. Model initialization significantly impacts performance, with its effect being particularly pronounced when training on the full DC-1B dataset, as indicated by UniTok's substantial increase from 70.8% to 78.6% with CLIP pre-trained weights. However, this initialization benefit is observed to be marginal for both CDD-VT and UniTok when evaluated on the characteristics of the Resampled DC-1B.

Expanding the training data from 800M to the full 1.28B DataComp-1B dataset provides only limited gains for the randomly initialized model ($66.8\% \rightarrow 70.1\%$). This marginal improvement suggests that, beyond sheer volume, the constrained diversity and repeated sampling within the additional 480M images restrict their contribution to performance. Such behavior is consistent with the intrinsic performance bound of pipelines employing solely a contrastive visual encoder (cf. "Upper Bound" row). Accordingly, future work will prioritize enhancing CDD-VT by exploiting training corpora with greater diversity and higher quality.

### 4.2 QUANTITATIVE EVALUATION

**Image Reconstruction.** Table 2 compares between CDD-VT and several prominent visual tokenizers, *e.g.*, LlamaGen (Sun et al., 2024), Show-1 (Xie et al., 2024), Cosmos (Agarwal et al., 2025), and Open-MAGVIT2-I-PT (Luo et al., 2024). CDD-VT achieves competitive results thanks to DPA, which dynamically selects primitives to reduce information loss. As shown in Figure 3, this allows for higher-fidelity image reconstructions than UniTok, especially in preserving fine details and text.

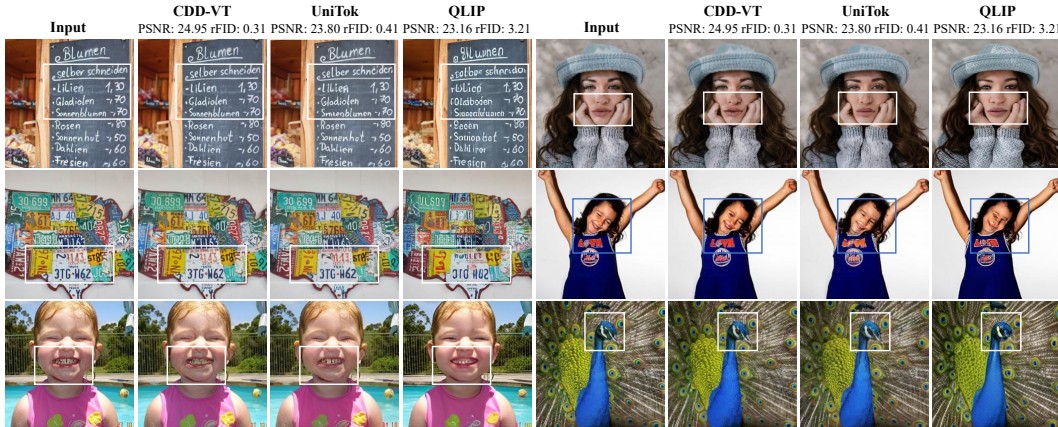

Figure 3: Comparisons of Image Reconstruction between UniTok (Ma et al., 2025), QLIP (Zhao et al., 2025), TokenFlow (Qu et al., 2025) and our CDD-VT. Here, FID and PSNR are evaluated on ImageNet 50k validation. CDD-VT shows superior detail preservation and perceptual fidelity over the competitors, *e.g.*, text on the map (Row 2) and blackboard (Row 1) is notably clearer and more faithful to inputs. Additional results can be found in Appendix C.

**Image-Text Retrieval.** We make comparisons with prominent competitors, divided into continuous tokenizers, *e.g.*, EVA02-B/16 (Fang et al., 2024) and CLIP-L/14 (Radford et al., 2021), and discrete tokenizers, *e.g.*, QLIP (Zhao et al., 2025) and UniTok (Ma et al., 2025). On the 800M Datacomp-1B subset, CDD-VT surpasses UniTok in accurately retrieving image-text pairs, showing strong cross-modal alignment. However, due to training constraints and information loss from vector quantization, it is not yet competitive with non-quantized CLIP models.

## 4.3 ABLATION STUDY

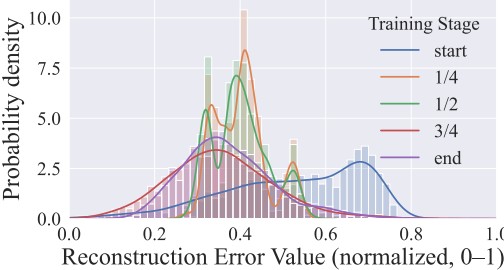

Figure 4: **Reconstruction Error during Training.**

Table 4: **Effect of the Warm-Up Training.** ✗ is no warm-up, *i.e.*, DPA and DQP are enabled from the start. ✓ indicates training with warm-up, *i.e.*, both modules are activated after the first quarter of training. Warm-up brings clear better results.

| Warm-Up Training | Accuracy | rFID↓ | LPIPS-VGG↓ |
|:---:|:---:|:---:|:---:|
| ✗ | 63.8 | 0.34 | 0.1206 |
| ✓ | 66.5 | 0.37 | 0.1410 |

**Warm-Up Training.** DPA is guided by reconstruction errors. However, these errors are unstable early in training, when the model has not yet learned effective reconstruction. As shown in Figure 4, reconstruction errors fluctuate heavily at the beginning but stabilize substantially by one-quarter of training. Table 4 ablates that activating adaptive quantization at this stage improves Top-1 accuracy, despite a slight drop in reconstruction. This suggests premature activation can hinder learning by over-allocating primitives in response to high initial reconstruction errors.

Table 5: **Left: DPA Effect on ImageNet 50K validation set. Right: Top-$K$ selection for the allocator $\mathcal{A}$.** All methods are trained on DC-800M, with 256 tokens and a downsampling ratio of 16.

| Method | DPA | Accuracy | rFID↓ | PSNR↑ | SSIM↑ | Top-$K$ | Accuracy | rFID↓ | PSNR↑ | SSIM↑ |
|:---|:---:|:---:|:---:|:---:|:---:|:---|:---:|:---:|:---:|:---:|
| Top-1 | ✗ | 65.8 | 0.41 | 24.17 | 0.69 | Top-500 | 65.5 | 0.94 | 23.74 | 0.70 |
| Top-10 | ✗ | 63.9 | 0.27 | 26.31 | 0.78 | Top-1000 | 66.5 | 0.37 | 23.93 | 0.71 |
| **CDD-VT** | ✓ | 66.5 | 0.37 | 23.93 | 0.71 | full | 64.7 | 0.44 | 24.23 | 0.71 |

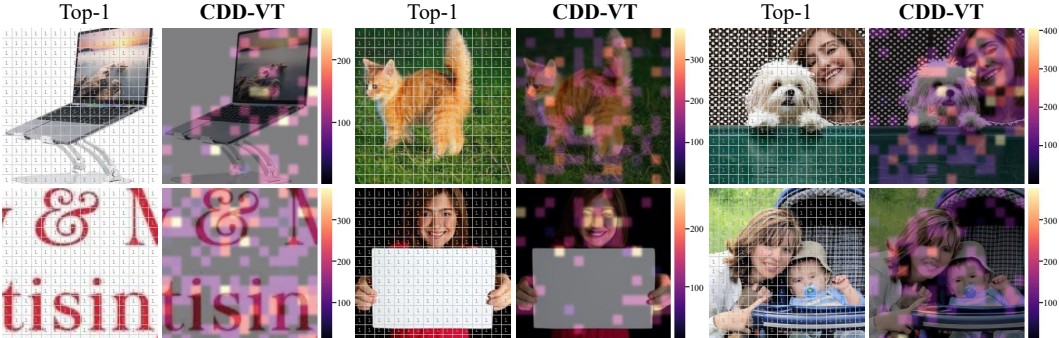

Figure 5: **DPA Effectiveness.** The Top-$K$ baseline (here $K$=1) assigns the fixed number of primitives to each patch, resulting in a uniform grid overlay. CDD-VT equipped with DPA produces an adaptive heatmap, focusing more primitives on areas with complex information. Color bars denote primitive count.

**Dynamic Primitive Allocator.** Table 5-Left ablates the effectiveness of DPA. A baseline using a fixed number of the Top-10 nearest primitives achieves outstanding reconstruction but poor text-image alignment. In contrast, our DPA successfully shows a better balance. Moreover, Figure 5 demonstrates visualizations of DPA, which adaptively focuses on areas with complex information, preserving fine-grained details in key regions while maintaining strong text-image alignment.

**Top-$K$ Selection for the allocator $\mathcal{A}$.** K is a crucial hyperparameter in the allocator $\mathcal{A}$, significantly impacts performance by dictating the pool size of nearest primitives. As shown in Table 5-Right, the best results is achieved with a Top-K value of 1000, which yields the highest accuracy and lowest reconstruction error. This demonstrates that selecting from a well-calibrated subset of the top-K nearest codes is more effective than either a smaller selection (Top-500, with a poorer or an unfiltered "full" selection, which sees a performance drop.

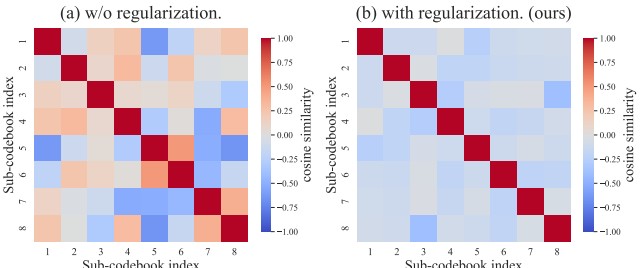

Figure 6: **Pairwise Similarity of Sub-Codebook Centroids.**

Table 6: **The Sub-Codebook Diversity Regularization** slightly improves Top-1 accuracy while markedly reducing rFID, demonstrating better understanding as well as generation.

| Codebook Diversity | Top-1 Accuracy | rFID↓ |
|:---:|:---:|:---:|
| ✗ | 66.1 | 0.43 |
| ✓ | 66.5 | 0.37 |

**Diverse Quantitative Primitives.** DQP introduces diversity regularization to penalize high similarity among sub-codebooks. Figure 6 visualizes the pairwise cosine-similarity of sub-codebook centroids. Learning with regularization, centroids are almost orthogonal, meaning sub-codebooks capture complementary information. Table 6 confirms the result quantitatively: regularization brings gains in Top-1 accuracy and reduces rFID, showing less information loss.

## 5 CONCLUSION

This work introduces CDD-VT, a continuous–discrete dualistic visual tokenizer designed to unify understanding and generation. It consists of an encoder, quantization, and decoder, with DQP and DPA being the core innovations of quantization. DQP fosters diversity of quantified primitives by ensuring distinct occupancy in the information space, which in turn improves the coverage of discrete primitives. DPA, on the other hand, mitigates information loss inherent during tokenization via dynamic primitive allocation. Experiments confirm that CDD-VT achieves performance on par with continuous tokenizers, yet being concise-elegant similar to discrete tokenizers.

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

## A    THE USE OF LARGE LANGUAGE MODELS (LLMS)

LLMs were employed to aid in polishing writing, enhancing linguistic clarity, refining the sentence structure of pre-existing ideas and analyses developed solely by human authors. LLMs' role was strictly limited to refining the linguistic expression of these pre-existing concepts, ensuring conciseness, grammatical correctness, and stylistic improvements without contributing to ideation, experimental design, data analysis, or the generation of any core scientific content or conclusions. This transparent disclosure aligns with our commitment to academic integrity and responsible research.

## B    LIMITATIONS

Unified understanding and generation models typically draw on diverse sources of training data, followed by extensive supervised fine-tuning. A significant challenge is the common practice of not publicly releasing proprietary training datasets, which hinders reproducibility and then comparative research. This opacity makes it difficult to ascertain the exact data distribution/scale behind many reported achievements.

We also clarify that CDD-VT tokenization was developed and evaluated independently. We did not, in fact, employ CDD-VT within the end-to-end training of Multimodal Large Language Models. Our focus here is solely on the robust design and performance of the vision tokenizer in its core functionalities of image representation and reconstruction, providing a strong foundation for future integration into comprehensive MLLM architectures.

## C  RECONSTRUCTION RESULTS

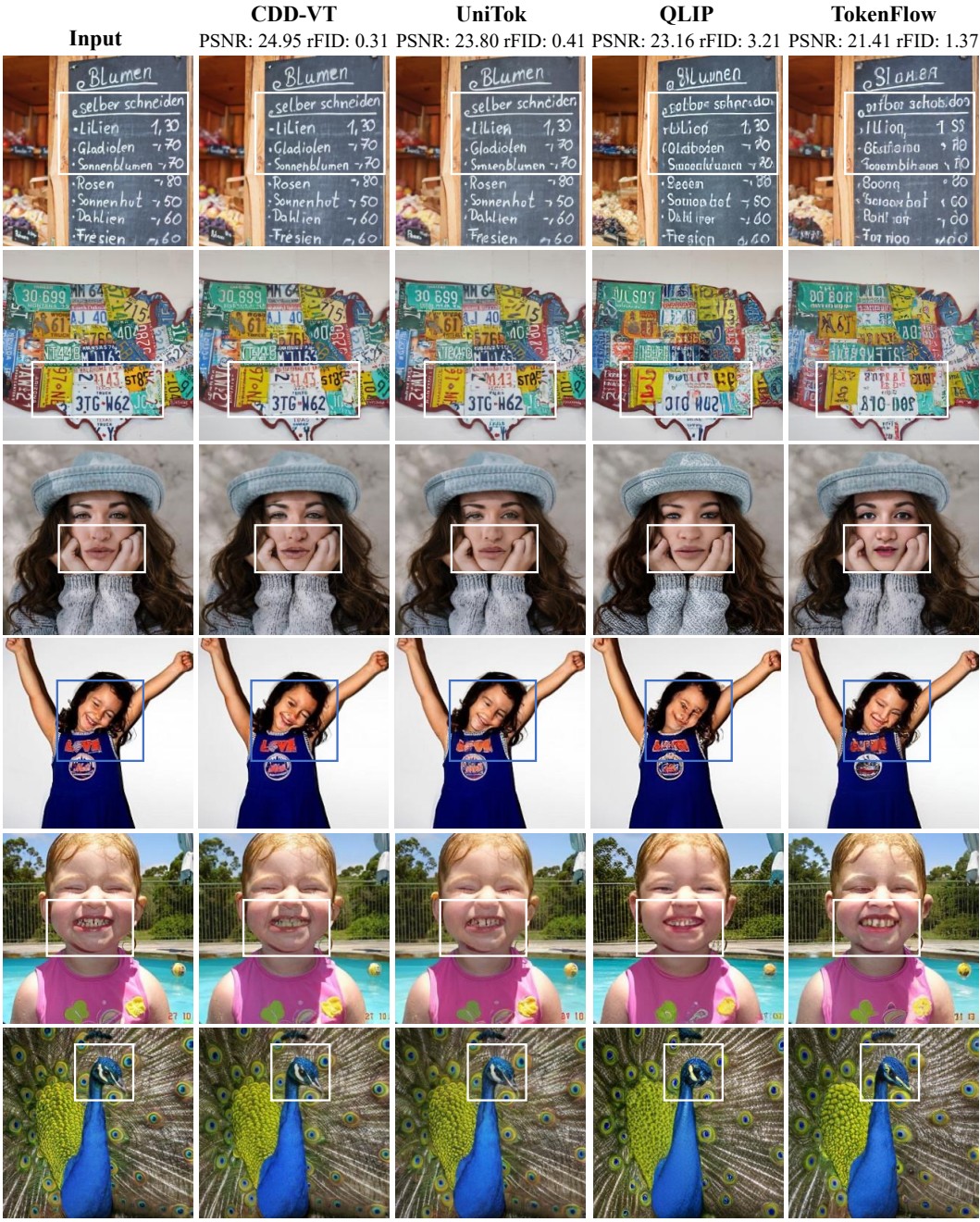

Figure 7: Comparisons of Image Reconstruction with detailed demonstrations: UniTok (Ma et al., 2025), QLIP (Zhao et al., 2025), TokenFlow (Qu et al., 2025), TokenFlow (Qu et al., 2025) and our CDD-VT. Here, FID and PSNR are evaluated on ImageNet 50k validation.

