# OpenReview forum: "Wave-Particle (Continuous–Discrete) Dualistic Visual Tokenization for Unified Understanding and Generation"
_ICLR.cc/2026/Conference — ICLR 2026 Conference Withdrawn Submission_

### Official Review · Reviewer_dDsP · 2025-10-15

**Soundness:** 2
**Presentation:** 3
**Contribution:** 2
**Rating:** 2
**Confidence:** 4

**Summary:**

This paper presents Continuous-Discrete Dualistic Visual Tokenizer (CDD-VT). Despite the name, the paper does not use a combination of discrete and continuous visual tokens to represent images. Instead, it uses a variable number of discrete visual tokens for each path (fewer for simple patches, more for complex patches). The authors use them for various downstream including image classification, image-text retrieval, and image reconstruction. The authors build a dynamic primitive allocator to decide how many tokens (primitives) to use for each patch, based on VAE-reconstruction errors. The authors compared their performance against UniTok and QLIP.

**Strengths:**

-	The proposed method dynamically assigns a different number of primitives to each image patch based on reconstruction error. This is interesting.
-	The diagrams and tables are clear.

**Weaknesses:**

-	The title and intro describe the proposed method as a hybrid continuous-discrete token representation. But actually, the “continuous” is only approximately by more discrete tokens. The claim of continuous-discrete is over-exaggerated. The real difference from prior works is the dynamically-determined, variable number of primitives for each image patch. This idea is similar to Images are Worth Variable Length of Representations by Mao et al.
-	Although similar in concept and attention-catching, using the word “Wave-Particle Duality” in the title might mislead readers. The wave concept could be confused with the wave-base positional encoding typical in Transformer models. Recommend renaming to avoid confusion.
-	Line 024, 223: It appears the authors meant “the number of primitives” rather than the “primitive number”.
-	Line 206: the authors did not explain clearly what a **sub**-codebook (why does stacking all primitives form a subcodebook?) and the rationale for splitting each embedding into M chunks. How does it increase diversity? Furthermore, this is no ablation study on this new logic.
-	The allocator in Section 3.3 relies on a VAE to compute construction error and use it to determine patch complexity. This VAE seems very independent from the rest of the model. Having it trained end-to-end might complicate things and require a warm-up, and still could misguide the model training. What if, you pretrain this VAE first, and freeze it during the training of the rest of the model? Would this save the overall training time?
-	The top 11 rows in Table 1 are from models trained using a different dataset, and thus, they are not comparable to the proposed methods. What is the purpose of putting them here? The results compared to UniTok seem marginal or degraded in accuracy.
-	Similarly, in Table 2/3, even for a zero-shot performance test, the training data should still be standardized.
-	In Table 4, when warm-up is used, accuracy increased, but FID and LIPIPS increased too, but the authors say “warm-up brings clear better results”.

**Questions:**

-	In Line 157, are H,W the size in pixels of each image patch or the number of patches?
-	In Line 184, what is the range of C_i? What does the O mean? Are the sizes of O and V different?
-	In Line 204, what do multi-element and multi-dimensional each refer to? What is different from prior works that use codebooks such as VQVAE?
-	In Line 206, if you cancatenate the codebook, the shape is V’xD’M, not V’xD’xM, right? This is from a stack() instead.
-	In Line 244, is the range 1/V’-1 or 1/V’ to 1?
-	In Line 269, the image encoder takes what into I?
-	In Table 5, what is the difference between CDD-VT and Top1/10? Top 1/10 are not methods but evaluation criteria, right? Is CDD-VT == Top-1000?
-	What is the difference between primitive and token?

---

### Official Review · Reviewer_dVqE · 2025-10-30

**Soundness:** 2
**Presentation:** 2
**Contribution:** 2
**Rating:** 2
**Confidence:** 3

**Summary:**

This paper, inspired by the integrated understanding-and-generation paradigm, unifies discrete and continuous tokenizers into a single framework, and proposes the Continuous-Discrete Dualistic Visual Tokenizer (CDD-VT), which dynamically selects between the two based on task complexity. The effectiveness of CDD-VT has been demonstrated on tasks including reconstruction, retrieval, and classification.

**Strengths:**

The paper proposes unifying discrete and continuous visual tokenization by adaptively using varying numbers of "image primitives," which may lead to some interesting phenomena and results.

The manuscript is clearly written and well-structured.

**Weaknesses:**

Experimental:

1. Although the paper is motivated by the paradigm of unified understanding and generation, it lacks validation on representative understanding and generation tasks. Instead, experiments are conducted on small-scale tasks and datasets, which limits the persuasiveness of the claims. The authors should supplement the evaluation with experiments on standard benchmarks for both visual understanding (e.g., image classification, VQA) and generation (e.g., image captioning, text-to-image retrieval) to fully demonstrate the effectiveness of the proposed unified framework.

2. The comparison with state-of-the-art methods is insufficient. Key recent approaches that bridge discrete and continuous representations—such as QLIP, TokLIP, and other contrastive learning frameworks like DINO, MoCo v3, and MAE—are not included in the experimental analysis. A comprehensive comparison with these baselines is necessary to position the proposed method within the current landscape and to justify its advantages.

3. The current experiments do not clearly isolate or demonstrate the benefit of merging discrete and continuous tokenization. It remains unclear whether the performance gain stems from the dualistic design itself, rather than from one of the individual components. The authors should provide ablation studies or direct comparisons against purely discrete and purely continuous variants under the same architectural and training settings to validate the added value of the hybrid approach.

[1] QLIP: Text-Aligned Visual Tokenization Unifies Auto-Regressive Multimodal Understanding and Generation, 2025.02
[2] TokLIP: Marry Visual Tokens to CLIP for Multimodal Comprehension and Generation, 2025.08

**Questions:**

See the Weaknesses

---

### Official Review · Reviewer_iVSb · 2025-10-31

**Soundness:** 3
**Presentation:** 3
**Contribution:** 2
**Rating:** 4
**Confidence:** 3

**Summary:**

The paper proposes CDD-VT, an adaptive visual tokenizer that bridges continuous and discrete tokenization through a “wave–particle” dual mechanism. Two key ideas are introduced: (1) DQP, which enforces diversity among multiple sub-codebooks, and (2) DPA, which allocates a variable number of primitives per image patch based on reconstruction error. Experiments on ImageNet, COCO, and Flickr30K show improved reconstruction and classification results over prior tokenizers like UniTok and QLIP.

**Strengths:**

1. **Clear motivation and good idea:**
The paper argues that the dichotomy between CT and DT limits multimodal model unification, and the proposed adaptive mechanism offers a reasonable trade-off solution. The DQP and DPA modules are simple yet effective.

2. **Strong reconstruction results:**
Quantitative results on ImageNet and COCO demonstrate measurable improvement in FID and PSNR over comparable discrete tokenizers (e.g., UniTok, QLIP). Qualitative reconstructions also look cleaner and more detailed.

3. **Solid ablation analysis:**
The ablations on DQP, DPA, warm-up strategy, and Top-K selection clearly show their impact on performance and training stability.

**Weaknesses:**

1. **Overclaim and Limited evaluation scope:**
The CDD-VT was claimed to be integrated into an end-to-end MLLM and was only tested as a tokenizer.  All results focus on image reconstruction, classification, and retrieval. No experiments show downstream generation or multimodal reasoning (e.g., text-to-image or image captioning).
2. **Dataset inconsistency.**
The experiments are conducted on a resampled DC-1B-R dataset instead of the official DataComp-1B, due to download issues according to the authors. This makes direct comparison with baselines questionable and reduces reproducibility. The authors should resolve the issue and redo the experiment on DC-1B for a proper comparison.
3. **Lack of efficiency analysis**
The authors didn't provide analysis of token efficiency. No runtime, memory, or throughput comparisons with baselines like UniTok or QLIP. Without this, it is hard to judge whether the proposed DPA/DQP modules offer a practical trade-off between performance and computational overhead.

**Questions:**

CDD-VT is a well-executed and thoughtful step toward unifying continuous and discrete tokenization. The adaptive primitive allocation is technically sound and empirically beneficial. However, the paper’s central unification claim is not fully substantiated, and dataset inconsistencies reduce reproducibility. Given these concerns, I'll not give a positive score at this moment.
If the authors can (1) rerun experiments on the official DataComp-1B and (2) demonstrate integration with an MLLM for generation tasks, I will raise my score

---

### Official Review · Reviewer_Wq2E · 2025-10-31

**Soundness:** 3
**Presentation:** 4
**Contribution:** 3
**Rating:** 4
**Confidence:** 3

**Summary:**

This paper proposes CDD-VT (Continuous–Discrete Dualistic Visual Tokenizer), a novel visual tokenizer designed to unify image understanding and generation within multimodal large language models (MLLMs). The key idea is to overcome the trade-off between discrete tokenizers, which are efficient but suffer from information loss, and continuous tokenizers, which preserve detail but are harder to integrate. CDD-VT introduces two components: DQP, which splits the codebook into multiple orthogonal sub-codebooks to encourage diversity and compositional expressiveness, and DPA, which dynamically allocates a variable number of primitives to each image patch based on content complexity. The resulting representations combine the strengths of discrete and continuous approaches and are mapped into the language token space via a lightweight MLP projector. This allows the MLLM to model visual and textual tokens jointly with a universal next-token loss. For generation, the model predicts continuous visual embeddings autoregressively, progressively denoising masked tokens. Experiments on image reconstruction, zero-shot classification, and cross-modal retrieval demonstrate significant improvements over existing tokenizers.

**Strengths:**

1. Novel tokenizer design: The paper proposes CDD-VT, a new visual tokenizer that effectively combines the advantages of discrete and continuous representations, addressing a fundamental limitation in existing visual tokenization approaches.

2. Well-motivated components: The introduction of DQP and DPA is conceptually sound and technically meaningful — DQP improves codebook diversity and compositionality, while DPA enables adaptive token allocation, enhancing the expressiveness of visual representations.

**Weaknesses:**

1. Recent works such as LaVIT [1] and DOVE [2] have also explored content-adaptive or variable-length visual tokenization, where more tokens are allocated to complex regions. While the proposed DPA module follows a similar motivation, the paper does not clearly discuss how its dynamic primitive allocation differs from these approaches. A more explicit comparison or clarification of the relationship would help better position the contribution of this work.
2. The main experimental results are not very competitive compared to strong baselines such as UniTok. In several metrics, UniTok outperforms the proposed method, and the reproduced baseline results presented in the paper lack sufficient credibility or detailed comparison. Without stronger quantitative improvements or a more rigorous baseline reproduction, it is difficult to assess the true effectiveness of the proposed approach.

[1] Jin, Yang, et al. "Unified Language-Vision Pretraining in LLM with Dynamic Discrete Visual Tokenization." ICLR. 2024.

[2] Mao, Lingjun, et al. "Images are Worth Variable Length of Representations." arXiv preprint arXiv:2506.03643 (2025).

**Questions:**

1. Could the authors elaborate on how the “continuous” aspect of CDD-VT is concretely represented in the model?
2. The paper claims that the DPA module allocates more primitives to semantically complex patches based on the assumption that larger VAE reconstruction errors correspond to higher information complexity. However, Figure 5 shows that many texture-sparse or simple patches are also assigned a large number of primitives. Could the authors provide experimental evidence or correlation analysis between the predicted R values, reconstruction errors, and actual image complexity to justify this assumption and clarify the mechanism behind DPA’s allocation behavior?
3. The paper states that 0.48B pairs were “resampled” to complete the DC-1B-R dataset. Could the authors clarify how this resampling was done and whether it affects data diversity?
4. Could the authors clarify what the “Upper Bound” mentioned in line 368 refers to?
5. In line 208, the paper mentions that the image is “split” and that selected primitives are combined into complete tokens. Could the authors clarify how this image-splitting process is implemented, and how the chosen primitives from DQP are aggregated to form a full patch-level token embedding?

---

### Note · Authors · 2025-12-03

I have read and agree with the venue's withdrawal policy on behalf of myself and my co-authors.